# Impact of Inpatient Unit Design Features on Overall Patient Experience and Perceived Room-Level Call Button Response

**DOI:** 10.3390/ijerph18189747

**Published:** 2021-09-16

**Authors:** Hui Cai, Francis Fullam, Lorissa MacAllister, Louis F. Fogg, Jeff Canar, Irwin Press, Connie Weissman, Olivia Velasquez

**Affiliations:** 1Institute of Health and Wellness Design, Department of Architecture, The University of Kansas, Lawrence, KS 66047, USA; huicai@ku.edu; 2Health Systems Management, Rush University, Chicago, IL 60612, USA; francis_fullam@rush.edu (F.F.); jeff_canar@rush.edu (J.C.); irwin_press@rush.edu (I.P.); 3Enviah, Grand Rapids, MI 49512, USA; 4College of Nursing, Rush University, Chicago, IL 60612, USA; louis_fogg@rush.edu; 5Department of Anthropology, University of Notre Dame, Notre Dame, IN 46556, USA; 6Rush University Medical Center, Chicago, IL 60612, USA; connie_weissman@rush.edu; 7HCA Healthcare, Brentwood, TN 37027, USA; olivia.velasquez@hcahealthcare.com

**Keywords:** nursing unit design, patient experience, patient satisfaction, response to call button, visibility, space syntax, HCAHPS, CAHPS, Press Ganey, orthopedic patients

## Abstract

This study explores the relationship between inpatient unit design and patient experience and how spatial features and visibility impact patients’ perception of staff responsiveness. The first part of this study is a retrospective pre–post and cross-sectional study evaluating the impacts of unit design on patient experience at the unit level. This study compares patient experiences based on Press Ganey and HCAHPS surveys in two orthopedic units (existing unit in Atrium building and new unit in Tower) with differing design features at Rush University Medical Center. The chi-square test results show that when moving from the old orthopedic unit to the new unit, almost all patient survey items related to patient experience showed statistically significant improvements. The second part of this study is a room level on the new unit. The ANOVA and Pearson correlation tests revealed that the visibility measure of metric step depth had significant impacts on patients’ perception of staff’s “promptness in responding to call button” and “help with toileting”. This study confirms that inpatient unit design plays a direct role in improvement for patient experience and should be considered as an important area of focus for future development.

## 1. Introduction

Patient’s perception of care—referred to as patient experience—is of great interest in the health care industry as it becomes more directly tied to how care quality is defined and the revenue of the health system providers [1,2,3]. In the 1980s, Press Ganey Associates LLC (Press Ganey) was founded to better the understanding of how patients perceive and rate their care with the development of standardized surveys. Press Ganey has grown to be a widely used vendor of patient surveys and offers client hospitals the ability to benchmark their patient experience against similar hospitals. As it became clearer that patient self-reported experiences were a critical subjective measure for hospital quality, the Centers for Medicare and Medicaid (CMS) published the first publicly reported survey called Hospital Consumer Assessment of Healthcare Providers and Systems (HCAHPS) in 2012. It covered the evaluation of care in multiple “domains of care” [4,5]. HCAHPS is used to calculate value-based incentive payments in the hospital value-based purchasing program, hence, it has implications for the financial performance of hospitals. The HCAHPS questionnaire is in the public domain and the Press Ganey survey is proprietary. Both surveys have evolved over time, but each stayed largely the same for the period of this study, 2010 through 2019.

Patient care occurs at many locations throughout the hospital; specifically for inpatients, their care typically occurs within their rooms. Therefore, understanding the impacts of inpatient nursing unit design, visual connectivity (syntax) and layout on patient experience is an important area of study for the industry [6,7]. A growing number of studies have demonstrated the importance of nursing unit design on patient experience [7,8]. It has been found that quality of care is related to patient experience in specific questions relating to nursing and patient communication, staff responsiveness, care transitions, and environment [9]. However, another study found that the new unit design could only improve the ratings for the facility-related questions, but not for nursing care quality and overall ratings of the hospital [10]. Moreover, most existing studies on patient experience and satisfaction remain at the hospital, patient type, or unit level. Determining a relationship in a dynamic environment is difficult; the variables used in the environment are fixed and the persons’ movement is fluid. To be able to isolate confounding variables and evaluate patient experience at the room level and link it to specific design features is important to inform future design [7]. Due to the dynamic nature of the study environment, the use of both the aggregated unit scores as well as the patient room-level scores can provide a more comprehensive understanding of the impacts of nursing unit layout on overall patients’ experience and how individual patient’s experience might vary by the location of patient rooms due to different visibility and accessibility from individual rooms to the nurse work stations. 

Among many aspects of patient experience, patients’ perception of nurses’ response to call buttons is an important factor [11,12]. “Call button” refers to the alert used by patients to receive medical attention as it aligned with the questions used in the Press Ganey and HCAHPS surveys. It should be noted that several references include the term “call light” in their works, and that it is synonymous with the term “call button”. The use of call buttons is considered to be one of the few means through which patients can exercise control over their care [13], as an important way to initiate patient–nurse communication [14], in terms of care and its delivery [15]. More importantly, responding to call buttons promptly can have implications for patient experience and safety. One study analyzed 108,882 instances of call button use collected from 27 units over a six-week period, and found that the seven most common reasons for call button use are: bathroom and bedpan assistance, intravenous line problems and pump alarms, accidental calls, miscellaneous, pain medication, nurse or certified nursing assistant needed, and repositioning or mobility assistance [16]. Similarly, Tzeng [17] identified the primary reasons for patient-initiated calls were for toileting assistance, pain medication, and intravenous line problems. Delayed response to call buttons can predict higher fall rates and injurious fall rates [18]. A higher call button use rate is associated with a less injurious fall rate [12,19]. When the average response time was longer, the patient experience score with “received help as soon as possible” was significantly lower [12,20]. The majority of these existing studies used hospital, or unit-month, or unit-week as the unit of analysis. No studies have evaluated the patients’ perception of response to call buttons at the room level.

Interventions to improve prompt response to call buttons mainly focus on nursing practice and staffing model. For example, the addition of a non-nurse staff position such as a unit assistant [11] or an unlicensed “patient service partner” to respond to call buttons [21,22] can decrease call button response time. Another intervention is to provide more frequent and consistent one-hour or two-hour nursing rounding, which was shown to be effective in significantly reducing the number of call button requests [23,24], improving patient experience, and reducing patient falls [16]. Others have proposed a call button intervention bundle by combining staff education, a patient-experience committee, purposeful rounding, and pod buddy assignments, which were shown to have positive impacts on patient experience with promptness to call buttons [25]. However, limited studies have investigated the role of the physical environment in improving prompt response to call buttons.

In the past two decades, there has been an increasing trend of shifting from a centralized nursing unit to a decentralized nursing unit (DNU) to keep nurses closer to patients, allow for better visual surveillance, and provide more prompt responses to patients’ needs. The DNU is defined as “one or more support spaces provided at multiple (two or more) locations on a physical inpatient unit” [26] (p. 57). A recent systematic review revealed a trend of positive change in patient experience after moving to some level of decentralization, based on patient survey data [6]. A few studies have investigated the impacts of DNU on call button use and patients’ perception of call button response using the Press Ganey survey. Friese et al. [27] collected the total number of call button uses on a monthly basis between July 2008 and May 2009, and found a decrease in the use of call buttons after moving to a decentralized nursing unit. Based on the analysis of the results of patient experience surveys for a six-month period, Guarascio-Howard and Malloch [28] found that the mean score for the question on staff’s promptness in response to call buttons is higher in a decentralized unit than centralized unit. Donahue [29] found that the decentralized unit with multi-hub pod (hybrid) design improved patients’ perception of nurses’ promptness in response to calls, attention to their personal needs, ability to keep the patient informed, and overall care rating in the Press Ganey survey as well as reduced incidences of patient complaints. Hua and colleagues [30] also found that the decentralized multi-hub (hybrid) design improved patients’ perceived nursing care quality including “promptness in response to call button,” “nurse attitude toward patient/family requests,” “being informed by the nurses,” and “pain control” based on the Press Ganey survey. However, a recent study also showed a slightly decreased rating of promptness in response to call buttons after moving into a hybrid decentralized nursing unit [31]. More evidence is needed regarding the impacts of nursing unit design, especially regarding the DNU layout on patients’ perception of nurses’ response to call buttons and patient experience.

More importantly, it is unclear whether visibility of patient units plays a role in staff’s response to call buttons. Previous studies have demonstrated the impacts of visibility on the effectiveness of patients’ visual surveillance [32], patient safety [33,34], nurses’ awareness of patients’ conditions [35,36], and nurses’ peer awareness and teamwork [35,37]. For instance, Cai and Zimring [36] found that the overall number of rooms that a nurse is aware of patient status is positively correlated to the visibility of the nurses’ assigned workstation (r = 0.715, *p* = 0.004). In other words, nurses tend to have a better understanding of patient status when there is a better visual connection between their workstations to the patient rooms and can potentially lead to more timely response. Hendrich et al. [38] used the space syntax method to analyze existing time and motion data in nursing units. They reveal that the nurse assigned to rooms with higher visual integration can lead to greater frequency of nurses’ visits to patient rooms and the nurse station. Both studies have used space syntax as a tool to quantitatively analyze visibility of the layout and specific locations in the unit (e.g., patient rooms, nurse stations). Space syntax is a theory developed by Bill Hiller and others to describe the social logic of space [39]. It has been recently increasingly applied in health care settings to evaluate the spatial configuration on patients’ and staff behavior and outcomes [40]. However, limited studies have explored the impacts of visibility of the layout on nurses’ promptness response to call button.

### Research Questions

This study aims to investigate the role the environment played in influencing patient experience of care. The project examined three research questions:Whether the decentralized inpatient unit design is associated with improved patient experience;Whether there is consistency between rooms in patient experience in decentralized inpatient room design or whether there are differences that may be associated with unit design;Whether the room’s visibility within the unit impacts patients’ perceptions of nurses and other staff’s response to call button requests and assistance with their needs.

## 2. Materials and Methods

### 2.1. Study Design

This study is based on a retrospective pre–post and cross-sectional study that evaluates the relationship between the nursing unit design and patients’ perception of nurse and staff responsiveness to patient needs for assistance. Institutional review board approval and informed consent were obtained from the Rush University Medical Center (RUMC, Approval code: 19101510-IRB01).

### 2.2. Settings

Rush University Medical Center (RUMC) is a 671-bed academic medical center in Chicago, Illinois. The unit selected for study was moved from an old building (the Atrium) to a new building (the Tower) in 2012. The Atrium building was built in 1979. It was based on a courtyard layout, with two nursing units on the north and south sides wrapping around two atriums (Figure 1). The nursing unit on each wing is based on a double-loaded corridor layout with a centralized nursing station in the middle of the unit. The selected unit for study (9 South) was on the 9th floor of the south wing with 32 single bedrooms. In the 13 East building, patients had the option of having their doors open or closed, but when doors were open, visibility was optimal for both patients and staff. The shortest distance from the nurse station to patient room (NS2 to 949S) is 11.2 ft., and the longest distance from the nurse station to patient room (NS2 to 938S) is 148.83 ft. 

The Tower was designed to optimize patient and family experience and improve the safety of patients and staff. It is a 14-story 830,000 square foot hospital building with 304 private acute and critical care beds. When the Tower opened in January of 2012, the orthopedic surgery unit (9 South) in the Atrium building was transferred to the new orthopedic unit on the east side of the 13th floor of the Tower (13 East). The patient type, care team, and nurse-to-patient ratio largely stayed the same between the units, so the comparison of Atrium 9 South prior-to-move and Tower 13 East after-move controls for these confounding variables. The 9 South unit had a mixed patient population, but the great majority were orthopedic surgery patients, and the remainder were general medicine patients (approximately 75% and 25%, respectively). The new 13 East unit was almost exclusively populated with orthopedic surgery patients (90%). In the analysis shown in this paper, only the orthopedic patients on both units are studied. The nurse-to-patient ratio for both units is based on one nurse per 4 patients on days during the business week; one nurse per 5 patients on evenings during the business week; and one nurse per 6 to 7 patients on the weekends.

13 East is located on the east side of the 13th floor with a butterfly-shaped floor plate. 13 East has 32 rooms and is divided into the north side (Rooms 1367–1382) and the south side (Rooms 1351–1366). It is based on a hybrid decentralized nursing unit design that combines 5 nurse stations with distributed nursing alcoves in front of each patient room. Patient assignments were associated with the locations of nurse stations. The rooms on the north side are covered by nurses located on Workstations #1, #2, #3, and the south rooms for nurses at Workstations #4 and #5 (Figure 2). For the north side, the shortest distance from the nurse station (#1, #2, and #3) to patient rooms is 45.44 ft., and the longest distance from the nurse station (#1, #2, and #3) to patient rooms is 102.09 ft. The average travel distance from the nurse station to all patient rooms is 67.92 ft. For the south side, the shortest distance from the nurse station (#4 and #5) to patient rooms is 51.53 ft., and the longest distance from the nurse station (#4 and #5) to patient rooms is 74.92 ft. The average travel distance from the nurse station to all patient rooms is 63.22 ft. The Tower had a new Rauland nurse call button system installed throughout the new facility.

Orthopedic units were selected for this study because they are relatively short-stay units, so patients are, for the most part, admitted and discharged from the same unit. The orthopedic patient population follows a predictable patient protocol for care. The convention for inpatient surveys is to send a survey to patients post-discharge and attribute their responses to their unit of discharge. Patients may stay on more than one unit in the course of their stay, but their responses about their patient experience are attributed to the last unit they were on before discharge. Because their responses are focused on the last unit they were in, this can cause some imprecision in terms of patient reporting and evaluating. Conducting the analysis on the orthopedic population eliminates this problem. In addition, patients on orthopedic units usually have a high call button use that is appropriate for room-level analysis [7].

### 2.3. Research Methods

#### 2.3.1. Tools to Assess Patient Experience

Patient experience with unit design issues at RUMC is assessed with secondary data analysis of the two commonly used inpatient questionnaires: HCAHPS and the Press Ganey inpatient surveys. Inpatients who are discharged to their homes are mailed a survey shortly after discharge by firms such as Press Ganey on behalf of medical centers.

Both questionnaires are an attempt to get feedback directly from patients about important aspects of their time in the hospital; however, the questionnaires differ in their philosophies and question formats. Press Ganey questions ask patients about their satisfaction and rating of care. The majority of the questions ask for ratings on a 5-point Likert scale: Very poor, Poor, Fair, Good, and Very good. The majority of HCAHPS questions ask patients about their experience of care and their report on how often something happened on a 4-point Likert scale: Never, Sometimes, Usually, and Always. The two questionnaires also differ in their organization and the issues covered. The Press Ganey questionnaire is organized under 11 sections with several questions under each (Admission, Room, Meals, Nurses, Tests and Treatment, Visitors and Family, Physicians, Discharge, Personal Issues, Emergency Department, Overall Assessment). The HCAHPS questionnaire in use at the time of this study had six composite subjective measures [41]: Communication with Nurses, Communication with Doctors, Responsiveness of Hospital Staff, Pain Management, Communication About Medicines, Discharge Information. In addition, there are two individual items related to different aspects of the hospital environment (cleanliness and quietness) and two global items (recommend the hospital and overall hospital rating). There is substantial overlap in content between the Press Ganey and HCAHPS questionnaires, and it is possible to cross-reference them around the framework of the meta-themes of care. The background demographic questions, screening questions (e.g., during the hospital stay did you need medicine for pain?) and the questions regarding the Emergency Department are excluded from this listing. See Table 1 for a listing of all the questions, arranged by meta-theme along with statistical testing comparing the aggregate differences between the Atrium unit and the Tower unit. While the two questionnaires use different response categories, there are two common ways data like these are analyzed—full-scale values and top-box. Both questionnaires use 4- and 5-point Likert scales for the most part and the HCAHPS survey has one 11-point Likert scale. The answers are assigned a numerical value, from low (least favorable response) to high (most favorable response). Improvements in the patient experience touched all areas of the patient experience related to the clinical care as seen in Table 1. Table 1 provides the detailed list of changes in patient ratings of the two units to show how large the improvements were across different questions.

In addition to full-scale values, we also used top-box scores for analysis. Top-box analysis dichotomizes responses on a scale to the “top box” (highest, most positive rating on a Likert scale) and responses that are not the highest. This is the convention used for the value-based payment system public reporting of the results for the HCAHPS results [42]. The table below shows examples of the different types of questions on the Press Ganey and HCAHPS surveys; the response categories the patients select from; the full-scale numerical values these responses are coded to; as well at the top-box conversion of the full-scale value responses (Table 2).

#### 2.3.2. Patient Experience of Care: Patient Experience and Nurses’ Response to Call Light

This retrospective, exploratory study measured patient experience at both the unit and room level using the Press Ganey survey results collected by RUMC. The unit-level analysis allowed us to compare the experience of the same type of clinical patients (orthopedic surgery) at the same medical center before and after moving to a new facility to look at the overall impact of the new facility design. It helped to compare the effectiveness of a centralized unit and a decentralized unit on patients’ perception of care quality and patient experience. The room-level analysis of individual patient’s PG response aggregated by room in the new decentralized unit compared patients’ perception of nurses’ response to call light, getting timely help, and assisting toileting. Since all patient rooms in the 13 East in the new Tower are same-handed with standardized design, the focus of the room-level analysis was on understanding whether the spatial configuration of the decentralized layout such as visibility of each patient room from nurse station affects individual patient’s perception of care quality and promptness to get help and nurses’ response to call.

The data cover patients discharged from 9 South Atrium between July 2010 and December 2011 (2801 Surveys) and 13 East between January 2012 and June 2019 (8143 surveys). The date range used for 9 South Atrium represent the last 17 months of the operation of the unit before it was transferred to the Tower. This provided enough patient survey cases to make comparisons at the unit level. It is not possible to study the room-level results for 9 South Atrium because room information was not recorded in the patient survey record until late 2011 so there is an insufficient number of patient survey responses that can be associated with specific rooms on the unit—room-level analysis is under powered. For the 13 East unit, we took an extended time period, 90 months, so that there are well over 100 survey responses per room.

Survey questions for this study included all standardized questions on both the Press Ganey and HCAHPS surveys relating to the assessment of care used during the study period. The survey questions and the analysis with the individual survey response directly coded to the room of discharge provide a clear ability to isolate the impact and effects of each unit design.

#### 2.3.3. Visibility Analysis

The visibility measures were analyzed based on the space syntax theory. Space syntax uses different techniques and measures to represent space as a relational spatial structure, including integration and step depth to measure visibility and accessibility. Integration is a global value based on the depth from one node to all other nodes in the system. In other words, it shows how well one space can be visible or accessible to the rest of the entire floor plan. Step depth (SD) describes the number of spatial depths of a given space from the origin, in other words, it represents the number of turns it takes to move from the origin to the destination.

The program, “Depthmap X” is used to conduct space syntax analysis and describe the patterns of visibility [37]. In addition to the integration as a generic measure of visibility, we adopted two relational metrics, visual step depth (VSD) and metric step depth (MSD) to describe specific spatial relationships between nurse stations and patient rooms. Step depth is a relational value and measures the number of turns (plus one) that needed to be traversed from the current location to see any other location within the plan [43]. The turn can be considered as a visual turn or a physical turn depending on whether the focus is on visual connectivity or physical accessibility. Every space that is directly visible from the selected origin is counted as one step away from that origin. In other words, step depth can represent the degree of accessibility between points of interests. For instance, if two rooms are connected with no turns, then the step depth is 1, if it requires one turn to get from one space to another, then the step depth is 2. The higher step depth represents fewer visual connections from the origin point to the destination in the Tower. As physical proximity is important for communication, we also include the measure of metric step depth, which is a weighted version of the step depth, considering the metric distance from one location to another [43].

In this study, we set each nurse station as the origin, and calculated the VSD and MSD from the origin nurse station to each patient room. Since the patient assignments were based on the wing and the workstations, the VSD and MSD of each patient room on the north wing (rooms 1367 to 1382) to nurse stations were calculated by the average VSD and MSD value to Nurse Stations #1, #2, and #3 (Figure 1 and Figure 2, Appendix A Table A1 and Table A2, and Figure A1); while the VSD and MSD of each patient room on the south wing (Rooms 1351–1366) to nurse stations were calculated based on the average value to Nurse Stations #4 and #5 (Figure 1 and Figure 2, Appendix A Table A1 and Table A2, and Figure A1). This value demonstrated how patients in each room perceived their visual connection and metric distance from the nurse stations that are in charge of their rooms. The study assumes that the patients’ doors were left open. There are no data on the extent to which the doors were open or closed.

#### 2.3.4. Statistical Analysis

All data were analyzed using Statistical Package for the Social Sciences (SPSS v 22.0, IBM, 1 Orchard Road, Armonk, NY, USA, 10504).

Descriptive statistics were used to analyze the data for the patient demographics and patient experience at the unit level (means and frequency distribution).

When comparing the patient samples on the two units, the chi-square test of independence was used to test for the association amongst categorical variables (gender, education, and race) and one-way between-subjects ANOVA test was used to test for group differences on continuous measures [44]. When analyzing the difference between units at an aggregate level for the patient survey responses of all the HCAHPS and Press Ganey survey questions, responses to the multipoint rating scale questions were dichotomized as “top box” score and “non-top box” score, as described earlier.

A chi-square test of independence was performed to compare the relationship between the unit designs (e.g., 9 South and 13 East) and the likelihood of selecting “top-box” in the patient experience.

When analyzing the difference between rooms on 13 East, the full values of the multi-point scales of patient experience surveys were used in the analysis. A one-way between-subjects ANOVA was conducted to compare the effect of room visibility in patients’ perception of staff’s promptness in responding to call and getting assistance such as help with toileting. The Pearson correlation coefficient was calculated between the patient survey responses and the space syntax variables at the room level.

## 3. Results

The results are reported in three parts. The first part compares the patients’ characteristics and patient experience data between the 9 South in the old Atrium building and the 13 East in the new Tower. The second part evaluates whether there are room-level differences in patient experience among rooms within 13 East. The third part analyzes the visibility measures of each patient room and whether the visibility measures are correlated with the room-level patient perception of staff’s promptness in responding to call button and providing assistance to help.

### 3.1. Comparing the Units

#### 3.1.1. Research Sample: Patient Demographics

The analysis of the demographics data of the orthopedic patients from these two units shows that the patient populations were similar (Table 3). The racial composition of the patients in these two units had no significant differences. Differences between patients’ gender, education, age, and the average length of stay (LOS) on each unit are small yet significant. The average age of patients in 9 South is 61.8, while those in 13 East are 63.7. The average (LOS) for an orthopedic patient declined by more than half a day between units (Table 4). This reflects the trend over time to get orthopedic patients up and walking as soon as possible after surgery and accelerate discharge.

#### 3.1.2. Aggregate Differences between Units

The chi-square analysis showed that virtually every aspect of the patient experience as measured on two widely used, standardized national patient surveys (i.e., Press Ganey and HCAHPS) showed significant improvement for orthopedic patients in the new Tower unit over the orthopedic patient experience in the Atrium. The top box score of every single question improved and these improvements were all statistically significant except the item related to the courtesy of the person who admitted the patient.

Not only are the improvements all significant but some are very large in absolute terms. The largest percentile change in patient evaluations of these two units was in the rating of the décor on the Press Ganey questionnaire, the top box value was 37.1% and on 13 East the top box value was 67.8%. This improvement in the raw score by 30.7 points equates to approximately a 60-percentile point improvement in performance on the Press Ganey benchmarking database of all hospital clients.

In addition to improved ratings of the facility-related items, there is also a substantial improvement in other aspects of patient experience related to care quality. The analysis shows that one of the questions on the HCAHPS survey designed as an overall evaluation of the hospital (rate this hospital during your stay), saw an improvement of 14.9% in the top box rating—from 69.2% for 9 South to 84.1% for 13 East. This equates to an approximate 40 percentile improvement on national benchmarking performance by CMS (HCAHPS Summary Analysis—HCAHPS Percentile Tables [45].

### 3.2. Differences between Rooms in the Tower

ANOVA did not reveal any significant differences between the rooms on the aspects of orthopedic patient experience in the Tower related to the responsiveness of the staff. The HCAHPS and Press Ganey questions on responsiveness to call buttons and toileting response are not significantly different by room.

The layout issues were further explored in the Tower unit by mapping the average ratings for these three variables on each room using heat maps below (Figure 3, Figure 4 and Figure 5). The colors go from yellow to red as the ratings go from high to low. These heat maps show that even though there were no statistically significant differences among various patient rooms, there might be patterns of differences not detected by ANOVA. Depending on the location of the rooms and the visual connections from the patient rooms to the nurse workstations, the pattern suggests that there may be some differences among rooms in terms of patients’ perception of staff’s promptness of responding to call buttons and getting help. Certain rooms such as Room 1369 and 1354 seem to be top performing among all rooms across these three measures. Additionally, when we compared the heat map for “promptness in responding to call button” and the heat map for “help with toileting as soon as you want”, it suggests that there are clusters of rooms with higher ratings in both items (Room 1354, 1359, 1364, 1369, 1380, and 1381).

### 3.3. Visibility and Patients’ Perception of Response to Call Buttons

#### 3.3.1. Visibility Analysis

The overall integration for 13 south units is 5.376, and the average integration for all patient rooms is 4.66 (SD = 0.69). The average visual step depth (VSD) from each patient room to all nurse stations is 3.07 (SD = 0.52) and the average metric step depth (MSD) is 1206.63 (SD = 259.67). The VSD and MSD were further analyzed based on the relationship between the room and the nurse stations that oversee these rooms (i.e., home-base nurse stations). The average VSD from patient rooms to the nurse stations that oversee these rooms on the south wing is 2.58 (SD = 0.66), and the average VSD to a home-base nurse station in the north wing is 2.56 (SD = 0.485). The average MSD from patient rooms to the nurse stations that oversee these rooms on the south wing is 640.54 (SD = 178.23) and the average MSD to a home-base nurse station in the north wing is 664.10 (SD = 150.95). The similarity of these values illustrates that there was no large difference between these two wings in terms of the visual connections from nurse stations to patient rooms in these two wings, even with travel distance and proximity considered. Detailed measures of VSD and MSD for each room in relation to each nurse station are reported in Appendix A.

#### 3.3.2. Visibility and Patient Experience with Call Button Response

The bivariate correlation on the call button response-related questions on the Press Ganey and HCAHPS surveys and visibility measures shows that there are moderate but significant correlations between visual connections between the patient room and nurse station and patients’ perception of nurses’ care (Table 5). The metric step depth from patient room to home-base nurse station is significantly correlated with the Press Ganey question on the promptness of the response to call buttons (Pearson r = −0.029, two-tailed significance 0.032) and significantly correlated with HCAHPS question on whether help toileting was received in time (Pearson r = −0.033, two-tailed significance 0.037).

## 4. Discussion

Results of the unit-level comparison showed that the new hybrid decentralized nursing unit design in the 13 East Tower had provided an improved patient experience in virtually all aspects when compared to the old 9 South Atrium. There were some anticipated improvements of ratings for the aspects of the patient experience related to the aesthetics of the facility, cleanliness, and noise level after moving from 9 South Atrium to 13 East Tower. However, the Tower design had statistically significant positive impacts on other aspects of patient experience which were not anticipated. Besides the “halo effect” of the new building, which is the tendency to positively rate an object on a particular attribute based on a generally positive impression of the object that has been well documented in psychometric and consumer research for some time [46], it appears that some design features in the 13 East in the new Tower are associated with improved patients’ ratings on care quality and overall experience.

As noted above, there was an improvement in the national HCAHPS percentile benchmarking performance on the overall rating of the hospital top-box scores by about 40 percentiles ranking among orthopedic patients moving between 9 South and 13 East, which is a marked improvement in relative performance. This level of improvement on this question and others can have a large impact on the hospital’s financial performance in the new value-based purchasing environment with increases in federal payments [2].

Compared to 9 South Atrium, the 13 East Tower also performed better in patients’ ratings on questions related to nurses’ speed to provide care and help such as “promptness in responding to call button (PG)” and “During this hospital stay, after you pressed the call button, how often did you get help as soon as you wanted it? (HCAHPS),” “How often did you get help in getting to the bathroom or in using a bedpan as soon as you,” “Nurses’ attitude toward your requests (PG), “and “How well your pain was controlled (PG). “These improvements might be attributed to the hybrid DNU layout that combined several nurse team-bases with distributed nursing alcoves in front of each patient room. The placement of nurses’ work areas in proximity to where care is delivered in combination with advanced communication technology allows nurses to be able to respond to patients’ needs in a more timely manner [10]. Compared to the centralized layout and long corridor for longer travel distance and limited visibility from the nurse station to patient rooms in 9 South Atrium, the decentralized layout of the 13 East Tower places clinical work areas and support spaces in closer proximity and better visual connections to patient care rooms which ultimately increases positive patient outcomes.

The room-level comparison in the new 13 East Tower unit did not reveal significant differences among rooms regarding patients’ perception of staff’s responsiveness to call buttons and request for help with toileting, which shows that the new layout supports a relatively higher level of consistency in patient experience regarding the responsiveness of the staff. With the decentralized hybrid layout, most patient rooms have similar proximity and visual connections to the nurse work areas, hence a more equity care experience. 

However, visualizing the room-level data as heat maps suggests that certain rooms with lower visual proximity to the nurse’s work areas seem to have consistently lower ratings in patient satisfaction, especially in outcomes associated with promptness of the response to call button and getting help toileting (Figure 3, Figure 4 and Figure 5). It suggests a greater importance of the placement of a room on the unit and its visual proximity to nurse stations on potential patient outcomes. The Pearson correlation further revealed that there was a statistically significant negative correlation between metric step depth (MSD) and patients’ perception of staff’s promptness of the response to call buttons (PG) and whether help toileting was received in time (HCAHPS). The lower MSD value indicates fewer visual turns and less travel distance between two points of interest. It shows that the distance-weighted visual connections from individual patient rooms to the corresponding nurse stations that are responsible for these rooms are critical [35,36] (Line 347). In other words, patients who stay in rooms with better visual connections and shorter distances from their home-base nurse stations tend to feel that they get faster response to call buttons and requests for help with toileting. This can potentially lead to better patient safety with fewer falls, based on previous research [19]. However, it needs to be further validated with the actual fall rate data.

## 5. Conclusions

### 5.1. Conclusions

This study provides important contributions to nursing unit design and justifies the value of an investment in evidence-based design in health care facilities. The results demonstrate the physical environment of the nursing units truly impacts patient experience and quality of care. In addition to the financial benefits to a better patient experience through the value-based purchasing program, there are other financial benefits of a positive experience. Fullam et al. [47] stated that excellent physician–patient communication can lower a medical center’s and physician’s risk of a patient lawsuit. Since the data cover 90 months after moving to the new Tower, this study reveals the long-term effect of design on patient experience, beyond the short-term “honeymoon” effect of a new building. The research provides new evidence to the field as it is different from the findings from a previous study in Johns Hopkins Hospital that claimed that the unit design only improved patient satisfaction on environment-related items but not on their perception of care quality from nurses, doctors, and other staff [10]. It appears that when unit design aligns with and supports care flow, operational model, and call button communication technology, it can lead to improved patient experience. The results also supported that the hybrid decentralized nursing unit can provide a more optimal patient experience, with patients’ perception of faster staff response to patients’ requests than centralized nursing units, which is consistent with previous studies [26,27,28,29].

This study also shows that visibility and proximity between patient rooms and nurse stations are important design characteristics that need to be carefully considered in the design, as they have a significant relation to patient perception of staff’s responsiveness to patients’ needs and their call button requests. Space syntax can be an effective tool to evaluate the design and inform the layout design before the actual construction of nursing units.

### 5.2. Limitations and Future Works

The following limitations are noted regarding the current study. Firstly, the results of this study were derived from nonequivalent groups. The background analysis that compared the demographics of the orthopedic populations on the two units found small but significant differences between the units in terms of the patient gender, education, length of stay, and age. These differences were not controlled for in subsequent analysis because of the small sizes of the differences. Given the extremely large effect size obtained and the ordering of the magnitude of those effects (from largest ‘pleasantness of the room decor’ to smallest ‘courtesy of the person who admitted you’) make it highly likely that room design is responsible for some of these substantial effects. It is also noted that there were not any operational changes in the care protocol for the patients so the research team was able to control the influence of the operational model as the confounding variable.

Secondly, the data for the old Atrium 9 South unit are not available at the room level. If the data were available, it might allow us to find room-level differences regarding patient experience and their perception of call button use. Future studies should investigate the room-level differences between centralized and decentralized nursing unit design.

The third limitation is that the average distance from nurse stations to patient rooms was not directly included as the independent variable for this study. Instead, the distance-weighted measure of metric step depth was used. Future studies can explore the impact of traveling distance on patients’ perception of nurses’ promptness to call.

The fourth limitation is that this study only reported patients’ perception of staff’s response to call buttons without including the actual call button use data. The nurse call button system used in RUMC documents each use of call buttons, indicating the frequency of call button use and the reason, which will be the subject of future publications. The research team plans to look at the relationship between unit design and the actual use of call buttons as a follow-up study.

Lastly, this paper focuses on one aspect of the patient experience (staff responsiveness) among many of the measures of patient experience. The research team plans to investigate the relationship between design features and other aspects of patient perceptions and some outcomes as well, i.e., patient perception of noise, cleanliness, and the design features which drive the aesthetic reaction to room design and fall rates. 

The Tower at RUMC provides a unique living laboratory to study the impact of design because RUMC has an integrated clinical database linked to patient-level survey data as well as extensive operational records regarding costs, labor, clinical outcomes, resource utilization, etc. The research team encourages other researchers in the evidence-based design field to take advantage of the patient survey data. These data points are available at most medical centers in the country to explore questions about the impact of facility design on patient experience and the use of resources.

Investments in new inpatient facilities are costly and long term. It should become routine to assess the impact of new facilities on clinical quality outcomes and cost of care. Facility design always strives to improve the quality of care and the tools are now available to include the assessment of the impact of the patient experience of care to be among the metrics to drive evidence-based design.

## Figures and Tables

**Figure 1 ijerph-18-09747-f001:**
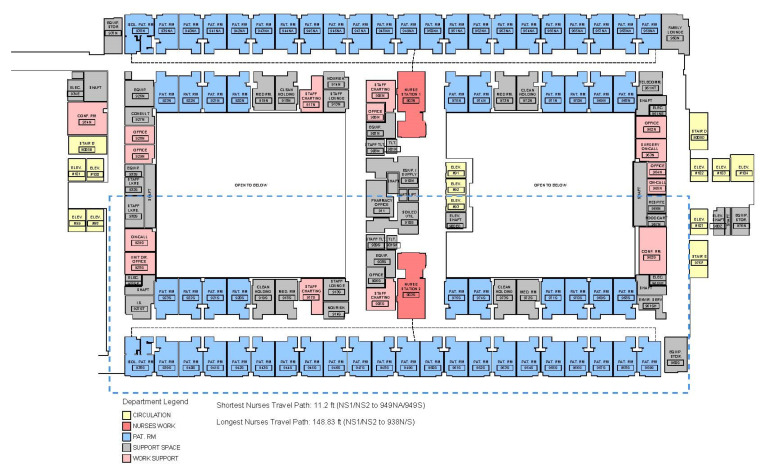
Floor plan of the 9 South in the Atrium building (Image Courtesy: RUMC, designed by Hansen Lind Meyer, plan re-drawn by the research team). Please see the dotted lines for area related to study and the drawing is not to scale.

**Figure 2 ijerph-18-09747-f002:**
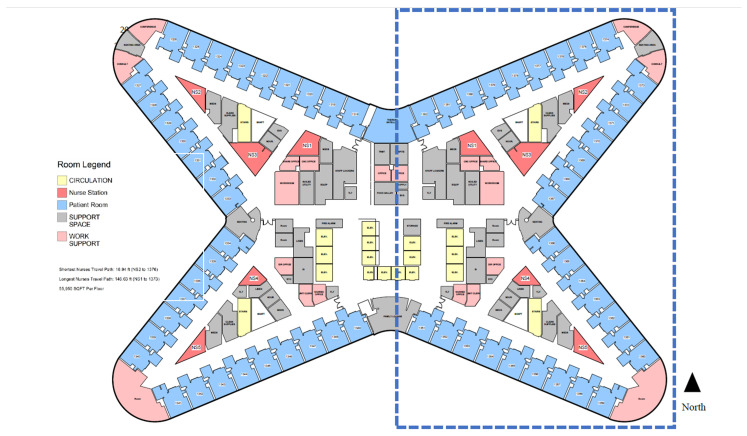
Floor plan of 13 East of new Tower (Image Courtesy: RUMC, designed by Perkins and Will, plan redrawn by the research team). Please see the dotted lines for area related to study and the drawing is not to scale.

**Figure 3 ijerph-18-09747-f003:**
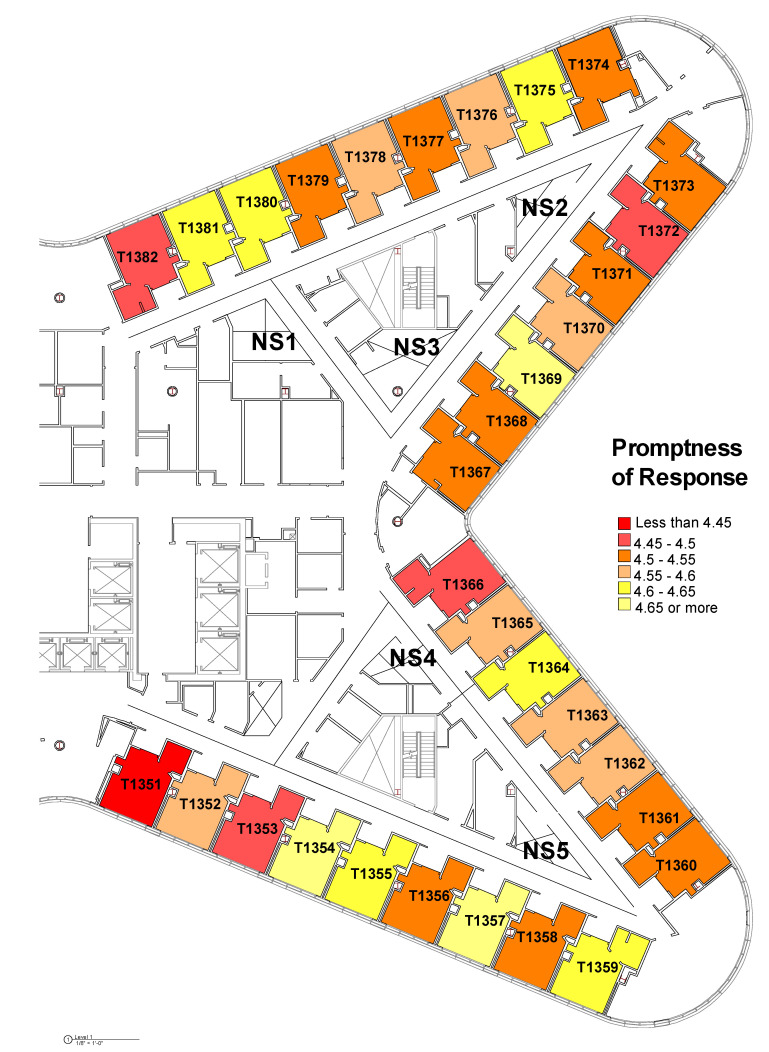
Heat map of room-level patients’ perception of staff’s “promptness in responding to call button” based on Press Ganey survey.

**Figure 4 ijerph-18-09747-f004:**
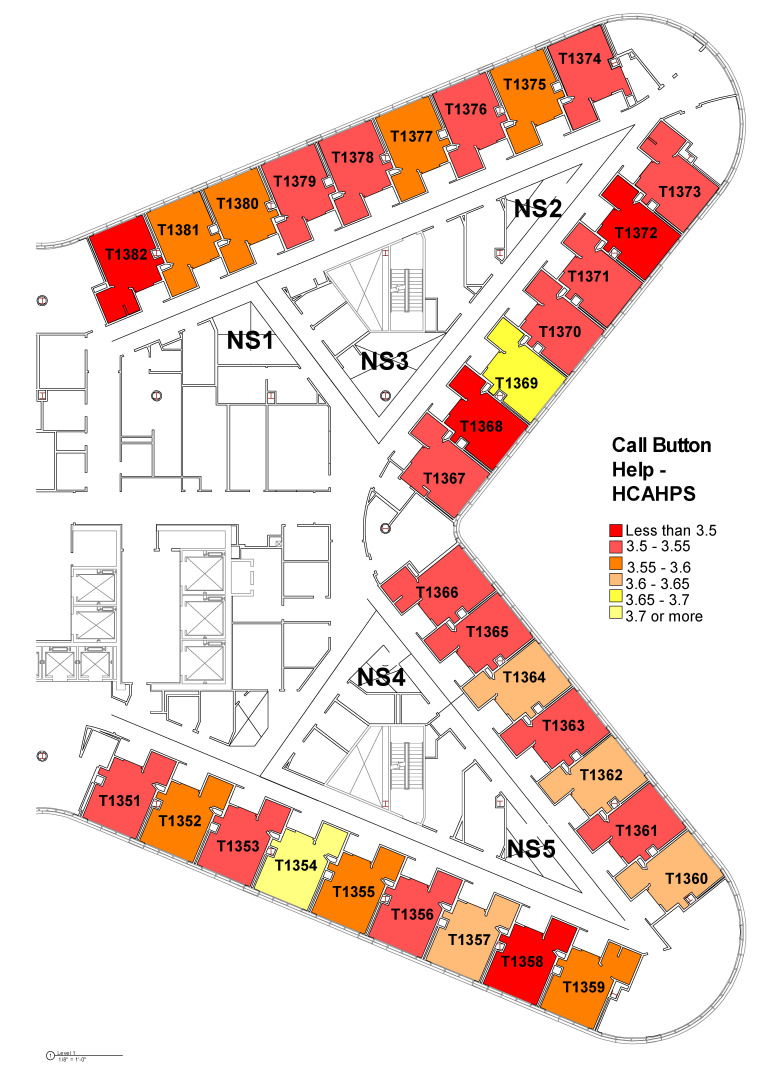
Heat map of room-level patients’ perception of staff’s response to “getting help as soon as you wanted after pressed call button” based on HCAHPS survey.

**Figure 5 ijerph-18-09747-f005:**
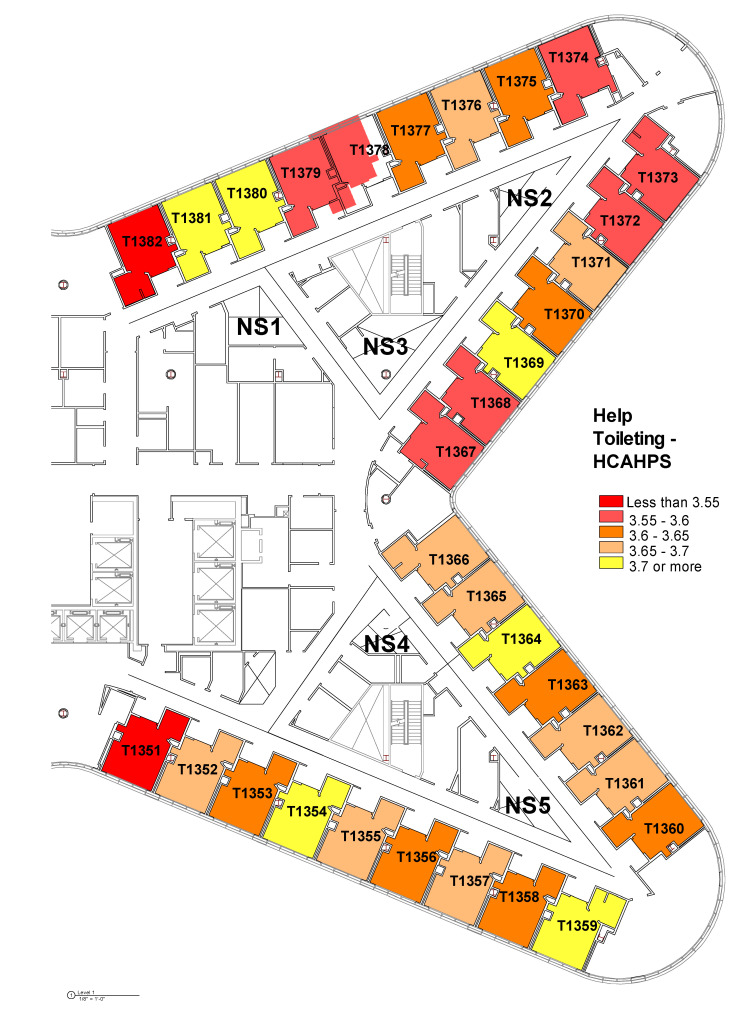
Heat map of room-level patients’ perception of “help toileting as soon as you wanted” based on HCAHPS survey.

**Table 1 ijerph-18-09747-t001:** Comprehensive list of all Press Ganey (PG) and HCAHPS questions, arranged by meta-theme along with statistical testing comparing the aggregate differences between 9 South and 13 East.

Meta-Theme	Question [Source]	9 South Atrium Top Box Score	9 South Atrium Number of Patient Surveys	13 East Tower Top Box Score	13 East Tower Number of Patient Surveys	Difference between Atrium and Tower Top Box Score	Chi-Square Value (All Significant at 0.00 Level, 2-Sided Asymptotic Test Unless Otherwise Noted)
Responsiveness	(Nurses) Promptness in responding to the call button [PG]	51%	1818	65.8%	5334	14.8%	126
	During the hospital stay, after you pressed the call button, how often did you get help as soon as you wanted? [HCAHPS]	49.9%	1708	63.2%	4871	13.3%	92.8
	How often did you get help in getting to the bathroom or using a bedpan as soon as you wanted? [HCAHPS]	61.6%	1161	73.7%	3925	12.1%	63.3
Nurses	Friendliness/courtesy of the nurses [PG]	74.7%	1943	83.3%	5642	8.6%	69.5
	Nurses’ attitude toward your requests [PG]	66.44%	1903	78.1%	5580	11.66%	104.5
	(Nurses) Amount of attention paid to your special or personal needs [PG]	63.1%	1900	75.5%	5560	12.4%	110
	How well the nurses kept you informed [PG]	61.5%	1918	74.8%	5597	13.3%	122.8
	Skill of the nurses [PG]	72.3%	1908	81.8%	5572	9.5%	78.5
	During this hospital stay, how often did nurses treat you with courtesy and respect? [HCAHPS]	81.2%	1967	89.5%	5759	8.3%	90.7
	During this hospital stay, how often did nurses listen carefully to you? [HCAHPS]	69%	1963	81.1%	5754	12.1%	125.7
	During this hospital stay, how often did nurses explain things in a way you could understand? [HCAHPS]	72%	1961	81.9%	5746	9.9%	88.4
Physicians	Time physician spent with you [PG]	41.3%	1866	50.4%	5438	9.1%	46.3
	Physician’s concern for your questions and worries [PG]	56%	1851	64.2%	5388	8.2%	39.6
	How well physician kept you informed [PG]	56%	1855	64.8%	5393	8.8%	45.5
	Friendliness/courtesy of the physician [PG]	66.5%	1868	74.8%	5446	8.3%	47.2
	Skill of physician [PG]	83.7%	1860	89.1%	5438	5.4%	38.3
	During this hospital stay, how often did doctors treat you with courtesy and respect? [HCAHPS]	85.3%	1948	90.8%	5716	5.5%	47.6
	During this hospital stay, how often did doctors listen carefully to you? [HCAHPS]	75.8%	1943	83%	5696	7.2%	49.5
	During this hospital stay, how often did doctors explain things in a way you could understand? [HCAHPS]	75.2%	1940	82.4%	5690	7.2%	47.9
Pain	How well your pain was controlled? [PG]	64.1%	1925	74.1%	5667	10%	70
	During this hospital stay, how often was your pain well-controlled? [HCAHPS]	64.9%	1864	70.9%	4293	6%	22.1
	During this hospital stay, how often did the hospital staff do everything they could to help you with your pain? [HCAHPS]	77.4%	1865	84.9%	4291	7.5%	50.6
Room	Pleasantness of room décor [PG]	37.1%	1926	67.8%	5616	30.7%	561
	Courtesy of person who cleaned your room [PG]	59%	1696	76%	5193	17%	182.5
	Room cleanliness [PG]	50.3%	1913	75.5%	5588	25.2%	422.4
	Room temperature [PG]	48.6%	1902	67.6%	5570	19%	218.9
	Noise level in and around your room [PG]	39.3%	1878	65.4%	5525	26.1%	395.7
	During this hospital stay, how often were your room and bathroom kept clean? [HCAHPS]	63.4%	1925	80.4%	5711	17%	228.9
	During the hospital stay, how often was the area around your room quiet at night [HCAHPS]	51.1%	1951	73.9%	5735	22.8%	350.1
Admission	Speed of admission process [PG]	74.8%	1946	77.9%	5606	3.1%	8.1
	Courtesy of the person who admitted you [PG]	84.2%	1805	85.8%	5348	1.6%	3 *
Discharge	Extent to which you felt ready to be discharged [PG]	61.1%	1925	71.1%	5659	10%	65.9
	Speed of discharge process [PG]	54.4%	1915	64.4%	5618	10%	60.4
Home Care	Instructions given about how to care for yourself at home [PG]	65.1%	1889	74.9%	5592	9.8%	68.4
	During this hospital stay, did doctors, nurses or other hospital staff talk with you about whether you would have the help you needed when you left the hospital? [HCAHPS]	43.2%	1386	49.7%	4143	6.5%	17.6
Overall Assessment	How well staff worked together to care for you [PG]	68.8%	1920	79.7%	5673	10.9%	95.2
	Likelihood of your recommending this hospital to others [PG]	73.3%	1937	85.3%	5684	12%	144.2
	Overall rating of care given at hospital [PG]	71.8%	1932	83.3%	5675	11.5%	120.8
	Using any number from 0 to 10, where 0 is the worst hospital possible and 10 is the best hospital possible, what number would you use to rate the hospital during your stay? [HCAHPS]	69.2%	1966	84.1%	5751	14.9%	204
	Would you recommend this hospital to your friends and family? [HCAHPS]	77.3%	1969	87.5%	5745	10.2%	120.1
Medication, Tests and Treatment	Waiting time for test and treatment [PG]	51.4%	1788	65.6%	5103	14.2%	113.1
	Explanations about what would happen during tests and treatments [PG]	58.9%	1773	71.7%	5090	12.8%	100.4
	Courtesy of the person who took your blood [PG]	69%	1900	78.9%	5430	9.9%	75
	Courtesy of person who started the IV [PG]	70.1%	1871	79.9%	5433	9.8%	76.4
	Before giving any new medicine, how often did hospital staff tell you what the medicine was for? [HCAHPS]	74.8%	1403	80.4%	4194	5.6%	19.7
	Before giving any new medicine, how often did hospital staff describe possible side effects in a way you could understand? [HCAHPS]	43.2%	1386	49.7%	4143	6.5%	17.6
Meals	Temperature of the food (cold foods cold, hot foods hot) [PG]	35.9%	1908	42.2%	5547	6.3%	23.6
	Quality of food [PG]	29.9%	1890	35.7%	5511	5.8%	21.1
	Courtesy of the person who served your food [PG]	61.9%	1880	67.4%	6474	5.5%	19.2
Visitors and Family	Accommodations and comfort for visitors [PG]	51.8%	1903	73.2%	5499	21.4%	296.4
	Staff attitude toward your visitors [PG]	66.8%	1874	81.3%	5463	14.5%	167.9
Personal Issues	Staff concern for your privacy [PG]	63%	1907	75.4%	5648	12.4%	109.7
	Degree to which staff addressed your emotional needs [PG]	57.7%	1782	69%	5351	11.3%	76.2
	Response to concerns/complaints made during your stay [PG]	58%	1733	70.3%	5191	12.3%	88.7
	Staff efforts to include you in decisions about your treatment [PG]	57.8%	1746	69.5%	5297	11.7%	80.6

* Not significant at the 0.00 level.

**Table 2 ijerph-18-09747-t002:** Listing of Press Gainey and HCAHPS survey example questions with full-scale numerical values.

(**a**) Press Ganey Question: Pleasantness of Room Décor numerical values.											
**Question (Press Ganey)**	**Pleasantness of Room** **Décor**										
Answer category	Very poor		Poor			Fair			Good		Very good
Numerical value assigned	1		2			3			4		5
Numerical value recoded as top box value	0		0			0			0		1
(**b**) HCAHPS Question: During This hospital stay, how often were your room and bathroom kept clean? Numerical values											
**Question (HCAHPS)**	**During This Hospital Stay, How Often Were Your Room and Bathroom Kept Clean?**										
Answer category	Never		Sometimes			Usually			Always		
Numerical value assigned	1		2			3			4		
Numerical value recoded as top box value	0		0			0			1		
(**c**) HCAHPS Question: What number would you use to rate this hospital during your stay numerical values											
**Question (HCAHPS)**	**Using Any Number from 0 to 10, Where 0 is the Worst Possible Hospital and 10 is the Best Hospital Possible, What Number Would You Use to Rate This Hospital during Your Stay?**										
Answer category	0—Worst hospital possible	1	2	3	4	5	6	7	8	9	10—Best hospital possible
Numerical value assigned	1	2	3	4	5	6	7	8	9	10	11
Numerical value recoded as top box value	0	0	0	0	0	0	0	0	0	1	1
(**d**) HCAHPS Question: Would you recommend this hospital to your friends and family? Numerical values											
**Question (HCAHPS)**	**Would You Recommend this Hospital to Your Friends and Family?**										
Answer categories	Definitely no		Probably no			Probably yes			Definitely yes		
Numerical value assigned	1		2			3			4		
Numerical value recoded as top box value	0		0			0			1		

**Table 3 ijerph-18-09747-t003:** Comparison of gender, education, and race between 9 South and 13 East.

	9 South Atrium	13 East Tower	Pearson Chi-Square Value	Significance
Male	40.8% (n = 811)	43.9% (n = 2486)	6.1	0.013
Female	59.2% (n = 1990)	56.1% (n = 5657)		
High school graduate or less	23.7% (n = 461)	19.6% (n = 1101)	22.5	0.000
Some college	30.6% (n = 594)	29.1% (n = 1639)		
College graduate	17.7% (n = 345)	20.5% (n = 1156)		
Graduate study	28.0% (n = 544)	30.8% (n = 1731)		
Race: white	83.6% (n = 1663)	82.9% (n = 2815)	0.55	0.459
Race: non-white	16.4% (n = 326)	17.1% (n = 994)		

**Table 4 ijerph-18-09747-t004:** Comparison of patient age and length of stay between 9 South and 13 East.

	9 South Atrium	13 East Tower	ANOVA F Value	Significance
Patient age	61.8 (n = 1990)	63.7 (n = 5774)	42.8	0.000
Length of stay	2.7 (n = 1941)	2.1 (n = 5809)	158.8	0.000

**Table 5 ijerph-18-09747-t005:** Pearson correlation analysis on visibility and patients’ perception of staff’s promptness of response to call light and assistance when requested.

Variable		Promptness of Response to Call Light (PG)	Call Button Help Soon as Wanted (HCAHPS)	Help Toileting as Soon as You Wanted (HCAHPS)
Visual Step Depth Integration Average Home-Base NS	Pearson r	−0.010	0.003	−0.024
	Significance (two-tailed)	0.450	0.830	0.140
	N	5333	4870	3924
Visual Integration Value	Pearson r	0.013	0.001	0.018
	Significance (2-trailed)	0.346	0.951	0.251
	N	5333	4870	3924
Metric Step Depth Average Home-Base NS	Pearson r	−0.029 *	−0.011	−0.033 *
	Significance (two-tailed)	0.032	0.443	0.037
	N	5333	4870	3924

* Correlation is significant at the 0.05 level (two-tailed significance).

## Data Availability

The data are not publicly available due to patient privacy.

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
