# Peer review of "Impact of Inpatient Unit Design Features on Overall Patient Experience and Perceived Room-Level Call Button Response"

_ijerph, 2021, doi:10.3390/ijerph18189747_

Round 1
Reviewer 1 Report
This is a solid study and the article is well organized. The introduction provided a good overview of existing studies on patients' experience and common indicators/metrics used in these studies. However, it was unclear why visibility was selected as a key factor in nurses' response to call. It would help to include literature that underlines the relationship between the visibility of hospital layout and staff's responses.
In the study, 9 south and 13 south represented centralized and decentralized inpatients. Are there any quantifiable features that illustrate/compare these two types of layouts, e.g., the average distance to the nurse station?
Space syntax was used to conduct visibility analysis in the Tower, but it was not introduced in detail in the method section. I would suggest that the authors explain the variables, e.g., integration, for readers who are not familiar with the software.
Reviewer 2 Report
Please see the attached word document.

Reviewer 3 Report
This is an interesting study on the impact of inpatient unit design on patient experience and perceived room-level call button response. The integration of the two parts of the study, focusing on the unit-level and the room-level, represents the richness and complexity of healthcare architectural design.
Here are a few comments to consider:
The study focuses on patient experience and their perception of care which is a subjective measure by nature. More explanations are needed to distinguish patient perception of care from the objective measure of the hospital quality of care. This topic is generally discussed in the limitation section but should be central in the introduction, discussion, and conclusion. The understanding that the study analysed the perception of care, which was impacted by the design of the unit, is crucial and should not be confused with the quality of care, which might have been the same in the old building (lines 447-448 for example).
The paper includes two studies on the unit-level and room-level. However, it is not clear why it is important to combine the two levels in the research. Please explain how one level impacts the other and what are the implications of studying the two together.
The response to call buttons is an objective, measurable performance indicator. Yet, the study did not include data on the number of calls, time of calls, waiting time, and co. in relation to the patient condition and location in the unit. Please explain why the study only referred to the patients perceived call button response and why it is an important measure although it is not objective.
Please add information about the technology used for the call button system in both cases. You state the name of the new nurse call button system in the Tower (lines 176-177), but more information is needed to understand how it is different from the previous system in the old building, and how do we know that the technological difference was not a confounding variable.
The abstract indicates the methods of Press Ganey, HCAHPS, Chi-square test, ANOVA, and Pearson correlation test, but they are explained only in the introduction and research methods. While some methods might be familiar (mainly in the US), others are less known and should be defined the first time they are mentioned in the paper. Also, please add references to the Chi-square test and ANOVA method (lines 257, 259).
Figure 1 and Figure 2 show the plans of the two buildings, while the research focuses on only parts of the floor plan (half?). I recommend showing the full floor plan and highlighting the area of the units/wings that were studied. If possible, please add the dimensions of the units/wings. Also, add the North sign to the plans to refer to the south wing in the Atrium building and the east side of the Tower building.
Space Syntax should be explained with references in the research methods in a section on tools to analyse the floor plan/architectural design, not just mentioned in the section on statistical analysis (line 272).
Section 2.2.1 jumps to 2.2.4 – sections 2.2.2 and 2.2.3 are missing.
Figure 3: It is difficult to read the legends. Please enlarge them and add indications for 3a, 3b, and 3c. Also, the numbers of the rooms are not readable, so it is impossible to know, for example, where Rooms 1369 and 1354 are located (lines 341-345) to tell if the two rooms are situated in the same place in the two wings. Please also highlight the number of the nurse stations in the drawing as a reference for Table 5.
Section 3.3. The visibility analysis based on the Space Syntax analysis must be explained before, in the methods section, including the definition of integration, VSD, and MSD.
Please provide more information on the visibility in the unit – are the doors constantly open? Can the patient see the nurse? Can the nurse see the patient in the room from the nurse station?
Table 5: Is it possible to illustrate the data on the floor plan for comparison to Figure 3? If not, I recommend placing Table 5 as an Appendix at the end of the paper. The table does not add to the understanding of the spatial analysis of the rooms in the unit (unless the reader is an expert in space syntax).
In the Discussion section, please provide more explanation of how the results demonstrate that the Tower's design was the reason for the positive impacts on other aspects of patient experience (lines 402-403). Also, the statement that visualising the room-level data as heat maps suggests something unique about the rooms in certain locations because their visual connections to nurse stations (lines 429-423) is not clear. Please explain how the results demonstrate this.
